# High Diagnostic Accuracy of RT-QuIC Assay in a Prospective Study of Patients with Suspected sCJD

**DOI:** 10.3390/ijms21030880

**Published:** 2020-01-30

**Authors:** Michele Fiorini, Giorgia Iselle, Daniela Perra, Matilde Bongianni, Stefano Capaldi, Luca Sacchetto, Sergio Ferrari, Aldo Mombello, Sarah Vascellari, Silvia Testi, Salvatore Monaco, Gianluigi Zanusso

**Affiliations:** 1Department of Neuroscience, Biomedicine and Movement, University of Verona, Piazzale L.A. Scuro, 10, 37134 Verona, Italy; isellegiorgia@gmail.com (G.I.); daniela.perra@univr.it (D.P.); matilde.bongianni@univr.it (M.B.); sergio.ferrari@aovr.veneto.it (S.F.); silvia.testi@univr.it (S.T.); salvatore.monaco@univr.it (S.M.); gianluigi.zanusso@univr.it (G.Z.); 2Department of Biotechnology, University of Verona, Cà Vignal 1, Strada Le Grazie 15, 37134 Verona, Italy; stefano.capaldi@univr.it; 3Surgery, Dentistry, Maternity and Infant Department, University of Verona, Piazzale L.A. Scuro, 10, 37134 Verona, Italy; luca.sacchetto@univr.it; 4Diagnostics and Public Health Department, University of Verona, Piazzale L.A. Scuro, 10, 37134 Verona, Italy; aldo.mombello@univr.it; 5Department of Biomedical Sciences, University of Cagliari, Cittadella Universitaria, 09042 Monserrato, Cagliari, Italy; svascellari@unica.it

**Keywords:** prion, CSF, RT-QuIC, sCJD, rapidly progressive dementia, biomarkers

## Abstract

The early and accurate in vivo diagnosis of sporadic Creutzfeldt–Jakob disease (sCJD) is essential in order to differentiate CJD from treatable rapidly progressive dementias. Diagnostic investigations supportive of clinical CJD diagnosis include magnetic resonance imaging (MRI), electroencephalogram (EEG), 14-3-3 protein detection, and/or real-time quaking-induced conversion (RT-QuIC) assay positivity in the cerebrospinal fluid (CSF) or in other tissues. The total CSF tau protein concentration has also been used in a clinical setting for improving the CJD diagnostic sensitivity and specificity. We analyzed 182 CSF samples and 42 olfactory mucosa (OM) brushings from patients suspected of having sCJD with rapidly progressive dementia (RPD), in order to determine the diagnostic accuracy of 14-3-3, the total tau protein, and the RT-QuIC assay. A probable and definite sCJD diagnosis was assessed in 102 patients. The RT-QuIC assay on the CSF samples showed a 100% specificity and a 96% sensitivity, significantly higher compared with 14-3-3 (84% sensitivity and 46% specificity) and tau (85% sensitivity and 70% specificity); however, the combination of RT-QuIC testing of the CSF and OM samples resulted in 100% sensitivity and specificity, proving a significantly higher accuracy of RT-QuIC compared with the surrogate biomarkers in the diagnostic setting of patients with RPD. Moreover, we showed that CSF blood contamination or high protein levels might interfere with RT-QuIC seeding. In conclusion, we provided further evidence that the inclusion of an RT-QuIC assay of the CSF and OM in the diagnostic criteria for sCJD has radically changed the clinical approach towards the diagnosis.

## 1. Introduction

Prion disorders are a group of rare, rapidly progressive, and fatal neurodegenerative diseases, which occur in sporadic, genetic, or acquired forms. They are caused by a conformational conversion of the normal cellular prion protein (PrP) into an abnormal misfolded pathological form, termed disease-associated PrP.

Disease-associated PrP can acquire different conformations, which, in combination with the status of methionine (M)/valine (V) polymorphism at codon 129 of the gene coding PrP, define the different molecular subtypes of sporadic Creutzfeldt–Jakob disease (sCJD), with distinctive clinical and neuropathological features.

sCJD commonly presents as rapidly progressive dementia (RPD), mimicking a number of potentially reversible forms, acronymized under the term VITAMINS [1], including, among others, neurodegenerative disorders, autoimmune encephalopathy, and infections [1]. In addition to clinical features, the diagnostic criteria for sCJD require the presence of specific changes in brain magnetic resonance imaging (MRI), electroencephalogram (EEG), and cerebrospinal fluid (CSF) biomarkers [2]. CSF 14-3-3 positivity has been included as a supportive test for sCJD diagnosis since 1998, although it is a non-specific surrogate marker of neuronal damage, which is also released intrathecally in other RPDs. However, despite this apparent lack of specificity, several studies have shown that, within an appropriate clinical setting, a positive 14-3-3 protein test can improve diagnostic accuracy [3]. The 14-3-3 expression is shown to be correlated with disease progression. In fact, 14-3-3 testing was found to be negative in the CSF samples of sCJD patients at preclinical stages and in clinico-pathological subgroups characterized by slow progression, while positive results have been observed in the CSF samples of the same patients in later stages of the disease [3,4].

Conversely, although not included in the diagnostic criteria for sCJD, the determination of the CSF levels of the microtubule-associated protein tau is helpful in diagnostic practice [3] and, in spite of a lower sensitivity, has a higher specificity than 14-3-3. CSF tau levels have been shown to strictly mirror disease progression, showing increasing values during the disease course [3,4].

Recently, in vitro prion amplification techniques, including the protein misfolding cyclic assay (PMCA) and real-time quaking-induced conversion (RT-QuIC) assays, have been set up for CJD diagnosis [5]. Intriguingly, RT-QuIC can detect a minute amount of pathologic prion protein in the CSF and olfactory mucosa (OM) from patients with sCJD, and it has been included in the revised diagnostic criteria [6]. As opposed to the earlier assay, named previous QuIC (PQ), showing a sensitivity of about 70% in CSF [7], the improved assay (IQ assay) of RT-QuIC has a sensitivity of about 95% [8]. We recently extended the RT-QuIC application to the olfactory mucosa samples of sCJD patients, reaching a high sensitivity and specificity, and we showed that the combination of CSF and OM testing by RT-QuIC can greatly improve the diagnostic accuracy in CJD diagnosis to nearly 100% [9].

The aim of the present study was to compare the diagnostic value of the old CSF biomarkers, 14-3-3 and tau, and the RT-QuIC amplification assay in sCJD. We evaluated prospectively the CSF 14-3-3 positivity, tau levels, and CSF RT-QuIC in 182 patients with possible sCJD, in addition to OM testing in 42 patients. We assessed the utility and the diagnostic value of each single marker in sCJD diagnosis.

## 2. Results

### 2.1. CSF Biomarkers Results and Diagnosis in RPD Patients

A total of 182 CSF samples and 42 OM samples were analyzed from patients with RPD and suspected sCJD.

Among the CSF samples, 130 resulted 14-3-3 positive, 111 had tau levels >1300 pg/mL, and 98 were RT-QuIC positive, while among the 42 OM samples, 32 were RT-QuIC positive; the surrogate biomarkers and RT-QuIC testing in the CSF and OM of each sample are reported in Appendix A.

At follow-up, 102 patients were diagnosed with sCJD. In detail, 61 patients were diagnosed with definite sCJD (sample code #1 to #61) and 41 patients received a diagnosis of probable sCJD (sample code # 62 to #102; Appendix A); the sCJD demographic, clinical, biochemical, and instrumental data are summarized in Table 1.

Eighty patients (sample codes #103 to #182 in Appendix A), hereafter referred to as “non-CJD”, received alternative diagnoses (shown in Table 2). Of note, in 11 cases, the brain tissues, obtained at autopsy, were analyzed by immunoblot and resulted in being negative for PK resistant disease-associated PrP (PrP^Sc^). The other 69 patients in the non-CJD group comprised cases with an alternative clinical diagnosis (e.g., strongly supported by neuroradiological and/or laboratory findings) or showed a clinical evolution incompatible with a prion disease (e.g., improvement or stabilization at follow-up).

The CSF and OM findings for the pathological and negative groups are reported in Table 3. The specificity of RT-QuIC was confirmed to be 100% both in CFS and OM, while by coupling the CSF and OM RT-QuIC, we obtained 61/61 and 41/41 positive results in the definite and probable sCJD groups, respectively. On the other hand, 14-3-3 and tau showed 43 and 24 false positives, respectively, in the non-CJD group.

### 2.2. Diagnostic Accuracy of CSF-RT-QuIC, OM RT-QuIC, and Surrogate Markers in Patients with sCJD Diagnosis

In subjects with a sCJD diagnosis, the RT-QuIC showed a sensitivity of 96% in CSF (95% CI, 92%–98%) and 91% (95% CI, 77%–97%) in OM. The specificity was 100% (CSF 95% CI, 97%–100%; OM 95% CI, 90%–100%), indicating the high reliability of the assay. Conversely, the 14-3-3 protein showed a sensitivity of 85% (95% CI, 79%–90%) and a specificity of 46% (95% CI, 39%–54%), and the tau sensitivity was 85% (95% CI, 79%–90%) and the specificity was 70% (95% CI, 63%–76%).

### 2.3. Atypical CSF Biomarkers in RT-QuIC Positive Samples

Nineteen cases with a probable (twelve) and definite (seven) sCJD diagnosis, with a CSF positive to RT-QuIC testing showed biomarker levels not orienting to sCJD clinical diagnosis (Table 4). In particular, 15 CSF samples were negative for 14-3-3, and 17 cases had tau <1300 pg/mL.

While in some CSF samples one or two biomarkers could be considered altered if not completely positive, three patients (#77, #80, and #93) had negative 14-3-3 and normal tau levels (<350 pg/mL). However, in all of these three samples, the RT-QuIC was positive both in the CSF and OM. Patient #100 was 14-3-3 negative and showed borderline normal tau level (373 pg/mL); this patient was found to be RT-QuIC positive in the CSF.

One CSF sample (#83) was 14-3-3 positive, but with low levels of tau.

Intriguingly, in the surrogate markers misleading group, we found the only three sCJD subjects with negative OM RT-QuIC (#57, #97, and #96); they were 14-3-3 negative in two cases and positive in one case, while they all had tau levels under the diagnostic cut-off of 1300 pg/mL.

### 2.4. Negative CSF RT-QuIC in Patient with Classical sCJD

Four patients displayed a typical clinical presentation and evolution with a disease duration of about two months, clinically classified as classical sCJD. The supporting investigations showed an EEG with periodic sharp wave complexes, and a hyperintense signal in the basal ganglia and cortical ribbon in the diffusion weighted imaging (DWI) was observed in the MRI. CSF was 14-3-3 positive and had tau levels higher than 1300 pg/mL. Unexpectedly, RT-QuIC was negative in CSF and, as a diagnostic confirmation, the patients who underwent nasal brushing and RT-QuIC in the OM sample had positive results. Three cases were autopsy confirmed (Table 5).

These CSF samples showed increased levels of total protein (>45 mg/dL) and albumin (>35 mg/dL); sample #33 was also blood contaminated. To determine if any interference for RT-QuIC by the CSF matrix effect or by pathological PrP in itself, in a dose dependent way, was present in the CSF samples, a RT-QuIC analysis was repeated on neat, 1:10-, and 1:20-fold diluted CSF in PBS. Unfortunately, only CSF samples #33 and #42 were available after the first CSF biomarker screening. As reported in Figure 1, the diluted samples indicated positive results in the RT-QuIC analysis, showing an increasing Thioflavin T (ThT) signal starting before 15 h, as for the positive undiluted control.

### 2.5. Negative OM RT-QuIC in sCJD

To prove if in the sCJD RT-QuIC negative OM samples the same interferences effects as for CSF could be avoided by sample dilution, we tested these samples at a 1:5, 1:10, 1:20, and 1:50 dilution. However, no seeding reaction was observed after the dilution of the OM samples.

## 3. Discussion

The RT-QuIC assay has already been shown to have a high diagnostic value, which is independent of the disease evolution [4]. In this study, any information about sCJD clinical disease progression or molecular typing cannot be clearly obtained by the main RT-QuIC assay parameters, such as the lag phase or the slope of the ThT signal.

In particular, the prion strain type of disease-associated PrP did not influence the RT-QuIC seeding reactions or the diagnostic accuracy. In particular, we did not observe differences of RT-QuIC sensitivity among CJD subtypes, as a variability in the RT-QuIC reactions was observed in the samples belonging to the same CJD molecular subtype. In addition, the use of a specific PrP substrate to improve the RT-QuIC sensitivity among sporadic CJD subtypes was not demonstrated, but we showed that we reached 100% sensitivity when CSF is coupled with OM brushing.

Conversely, the surrogate biomarkers have a potential prognostic value related to their levels, as they are found to be normal in sCJD with a long disease duration [4], and their expression has been demonstrated to be correlated with the molecular subtypes [10].

In this prospective study, we found a high overall diagnostic accuracy for CSF RT-QuIC, in line with previous studies, [8,9,11,12], and we showed that the combination of CSF and OM RT-QUIC assays provided a final diagnostic accuracy of 100%, confirming our previous findings in a different cohort of subjects [9]. We further compared the diagnostic accuracy of prion detection by RT-QuIC in the CSF and OM samples to surrogate biomarkers evaluation, and we demonstrated the higher diagnostic accuracy of RT-QuIC than CSF determination of 14-3-3 and tau.

We previously showed that 14-3-3 was positive in the CSF of neuroleptic-treated patients [13], as well as other neurologic and inflammatory diseases, such as Guillain Barrè syndrome [14], post-polio syndrome [15], acute myelitis [16], and multiple sclerosis [17]. Most of these CSF samples had normal tau levels, likely because the degree of neuronal or axonal damage was not severe enough to determine an increase in released tau protein in the CSF [13]. As is widely known, tau levels increase consistently in other neurological disorders, such as encephalitis [18], stroke [19], and brain tumors [20], or in subarachnoid hemorrhage [21]. Taken together, these findings indicate that both 14-3-3 and tau are highly sensitive, but have a low specificity. In contrast, in a clinical context, the 100% specificity of RT-QuIC is the most relevant issue for the patient.

Although prion diseases are characterized by important neuronal damage, we found several CSF samples with negative 14-3-3 and with normal tau levels, resulting in a final sensitivity of 84% and 85% for the 14-3-3 and tau assays, respectively. These results are far from what we reported in our previous study on 127 definite sCJD cases, where we showed a diagnostic sensitivity of 96% for CSF 14-3-3 detection and 92% tau determination [10]. We could speculate that in the present study, the reduced sensitivity may be as a result of a bias in patient selection, which was less screened.

In three patients, the CSF samples were negative for both 14-3-3 and tau and were completely non-diagnostic. However, an MRI scan showed a hyperintense signal in the cortical ribbon at DWI, which progressed to diffused brain involvement in the subsequent MRI scans. The CSF was RT-QuIC positive in all of these subjects.

These findings indicate that in presence of normal biomarkers, such as a negative 14-3-3 and normal tau, sCJD diagnosis might not be excluded. In these cases, the early diagnosis of a prion disorder is important for clinicians in order to avoid additional and unnecessary diagnostic investigations, a relatively common condition before the introduction of RT-QuIC. Besides a reduction in the cost of the diagnosis, a precise diagnosis has relevant ethical implications for both the patient and the family. Paradoxically, four sCJD cases with a typical disease phenotype were RT-QuIC negative in the CSF, but this test was not relevant for the diagnosis.

In the four cases with negative RT-QuIC in the CSF, 14-3-3 was positive and tau was >1300 pg/mL. Among them, the three definite sCJD cases showed onset and clinical signs, MRI, and EEG, suggesting typical sCJD, and had similar short duration. However, the negative RT-QuIC result in the CSF was likely related to blood contamination or a high protein concentration, which might interfere with the prion seeding reactions. In fact, when these samples were diluted, a positive response to the RT-QuIC assay was obtained. Inhibition in ThT fluorescence by blood has been recently confirmed by Foutz et al., showing a direct interference by increasing the blood concentration in th RT-QuIC reaction [11,22]; in particular, ThT is known to interact with several molecules, such as serum albumins [23]. We cannot exclude that the CSFs were blood contaminated, and only blood cells, not serum proteins, could be eliminated by centrifugation. Although, at the moment, the number of CSF samples that become positive after dilution is not consistent, a retesting of CSF false negative RT-QuIC in diluted samples may be a useful approach to further rise the sensitivity of the assay.

Nevertheless, these sCJD cases with an apparent negative CSF RT-QuIC received a diagnosis of CJD, because the OM samples of the same patients have been tested positive. These data confirm that OM RT-QuIC may, however, widely fill the gap in the case of negative CSF RT-QuIC, corroborating what we previously reported [7,9]. Despite the fact that the OM RT-QuIC sensitivity and specificity was in line with previous findings [7,9], we found the three negative OM RT-QuIC among the sCJD cases, and we can argue improper OM sampling.

In conclusion, we showed that, by exploiting its extremely high sensitivity and specificity, CSF RT-QuIC, alone or in combination with OM RT-QuIC, can give an early and accurate in vivo diagnosis of sCJD, avoiding dangerous false negative results by surrogate biomarkers.

Moreover, considering that, in less than one week, RT-QuIC gives such sensitive and specific results, and in most cases at the early stages of the disease, it needs to be underlined how a first screening by RT-QuIC can help in a reduction of the cost of diagnosis of a prion disorder, avoiding long and expensive, less accurate instrumental and molecular investigations.

Finally, given the absolute specificity of RT-QuIC, a positive result may be considered as a definite diagnosis, and this implication may lead to considering and evaluating if autopsy is really necessary. In negative RT-QuIC cases, however, very few cases of sporadic or variant CJD [5] cannot be excluded, and post-mortem biochemical or pathological confirmation is mandatory.

## 4. Materials and Methods

### 4.1. Patients and Samples

This study included 182 patients presenting with RPD, referred to the Neuropathology Laboratory in Azienda Ospedaliera Universitaria Integrata of Verona, Verona, Italy, because of the clinical suspicion of CJD, between January 2015 and December 2017. The final diagnosis of these patients was established at death or when a definite alternative diagnosis was available. The diagnosis of human prion disease was made according to internationally established criteria [6].

The ethical committees of the Azienda Ospedaliera Universitaria Integrata of Verona, Italy, approved the study (Prot. n. 28917 15 June 2012). Written informed consent for participation in the research was obtained in accordance with the Declaration of Helsinki (1964–2008) [24], and the Additional Protocol on the Convention of Human Rights and Biomedicine Concerning Biomedical Research (2005) [25].

One hundred and eighty-two CSF samples were collected by lumbar puncture, were aliquoted as soon as collected, and were stored at −80 °C.

Forty-two OM specimens were obtained by nasal brushing and processed as previously described [9]. Out of the 42 samples, 31 OM samples were obtained at the time of lumbar puncture, while the other 11 OM samples were obtained from patients with negative CSF RT-QuIC, in order to confirm CJD diagnosis [9].

Written informed consent from each patient or their representatives was obtained before each CSF and OM sampling.

### 4.2. Genetic PRNP Analysis and Codon 129 Determination

After informed consent, genetic analysis was carried out on the DNA extracted from the blood specimens to exclude mutations of the gene *PRNP* coding for PrP and to determine M/V polymorphism at codon 129 for molecular classification.

### 4.3. CSF Surrogate Biomarkers Analysis

CSF 14-3-3 was detected by immunoblot. For each sample, the equivalent of 25 μL of CSF was loaded onto a 13% polyacrylamide gel and transferred to polyvinylidene fluoride membranes, as previously described [26]. Thee membranes were incubated with anti-14-3-3 rabbit polyclonal antibody (Santa Cruz Biotechnology, Dallas, TX, USA), and immunoblot was revealed using an enhanced chemiluminescence system. The 14-3-3 testing was judged to be positive (+) or negative (−) compared with the positive control.

The CSF tau protein concentrations were measured in duplicate by sandwich ELISA, using the INNOTEST^®^ hTAU Ag ELISA kit (Fujirebio Europe, Gent, Belgium), according to the manufacturer’s instructions. The absorbance values were obtained with a microplate reader and the tau concentrations were estimated from standard curves made for each assay.

### 4.4. RT-QuIC Analysis

The recombinant PrP substrates were prepared, as previously described [7,9]. The RT-QuIC assays were performed, as reported previously, for CSF improved RT-QuIC (IQ-CSF) and OM in a plate reader (FLUOstar Omega; BMG LABTECH, Ortenberg, Germany), with cycles of 90 s of shaking (900 rpm, double-orbital) and 30 s of rest throughout the incubation. For the CSF analysis, reactions were run at 55 °C with hamster recombinant PrP 90-231; twenty microliters of undiluted CSF were used per reaction well. For the olfactory mucosa analyses, the plates were incubated with hamster recombinant PrP 23-231 at 42 °C for 55 h. The thioflavin T (ThT) fluorescence measurements (mean excitation, 450 ± 10 nm; mean emission, 480 ±10 nm (bottom read)) were taken every 45 min. The sample findings were judged to be RT-QuIC positive using criteria, similar to those previously described for the RT-QuIC analyses of the OM and CSF specimens [7,9]. All of the CSF RT-QuIC determinations in this work can be identified as the previously described IQ-CSF.

### 4.5. Brain Samples and Proteinase K-Resistant Prion Immunoblot Analysis

To determine the definite sCJD diagnosis, samples from the frontal, occipital cortex, and cerebellum were collected from autopsied brains, to be tested for the presence of prion proteinase K-resistant fragment, as previously described [27]. Brain tissue samples were kept at −80 °C until use. The brain tissues were homogenized in nine volumes of lysis buffer (100 mM sodium chloride, 10 mM EDTA, 0.5% Nonidet P-40, 0.5% sodium deoxycholate, 10 mM Tris, and pH 7.4). The aliquots were digested with 50 μg/mL of proteinase K at 37 °C for 60 min; the samples were then separated by SDS-PAGE gels, and the proteins were transferred onto a PVDF membrane (Immobilon P, Millipore, Burlington, MA, USA). PrP^Sc^ was detected by a 3F4 anti-human PrP monoclonal antibody. The blots were developed with an enhanced chemiluminescence system (ECL; Amersham Biosciences, Little Chalfont, UK) and PrP visualized on autoradiographic films (Hyperfilm, Amersham Biosciences).

### 4.6. Statistical Analysis

Statistical comparisons of mean relative demographical data of different patient groups were performed by the t-test. The sensitivity, specificity, and relative 95% confidence intervals (CI) of the RT-QuIC, 14-3-3, and tau protein were calculated.

## Figures and Tables

**Figure 1 ijms-21-00880-f001:**
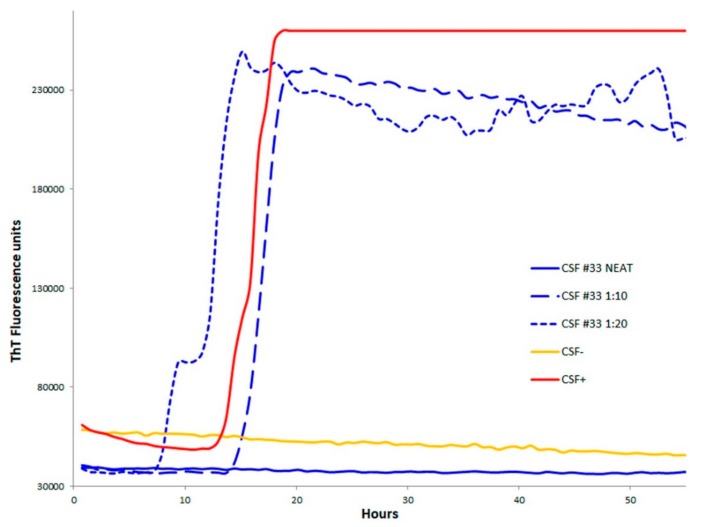
Patient #33 RT-QuIC analysis of neat and diluted (1:10 and 1:20). The analysis is also representative of CSF sample #41.

**Table 1 ijms-21-00880-t001:** Summary of probable and definite sporadic Creutzfeldt–Jakob disease (sCJD).

	Definite sCJD (*n* = 61)	Probable sCJD (*n* = 41)
Male/female	31/30	23/18
Age, in years (± SD)	69 (9) *	66 (8) *
Mean disease duration, in months (range)	5.5 (1–21) *	7 (2–35) *
Mean interval between onset and spinal tap, in months (range)	3 (1–13) *	5 (1–14) *
Type of PK resistant disase-associated PrP	T.1 *n* = 42; T.2 *n* = 16; T.1/2 *n* = 3	-
Typical EEG	22/56	10/29
Typical MRI	45/54 **	21/31 **
Diagnosis before CSF analysis	Possible CJD *n* = 12 Probable CJD *n* = 49	Possible CJD *n* = 14 Probable CJD *n* = 27
Diagnosis after CSF analysis	Possible CJD *n* = 0 Probable CJD *n* = 61	Possible CJD *n* = 0 Probable CJD *n* = 41

* *p* > 0.1; ** In non-typical MRI only one cortical area was affected. PrP—prion protein; SD—standard deviation; PK—proteinase K; EEG—electroencephalogram; MRI—magnetic resonance imaging; CSF—cerebrospinal fluid.

**Table 2 ijms-21-00880-t002:** Diagnostic categories non-CJD.

Clinical Diagnosis	Number of Cases
Alzheimer’s disease	34
Frontotemporal dementia	5
Lewy body dementia	3
Other neurodegenerative diseases	3 *
Encephalitis (infectious or autoimmune)	5
Psychiatric disease	2
Toxic/metabolic encephalopathies	5
Vascular dementia	8
Central Nervous System malignancy	2 **
Cause unknown (improvement at follow-up)	4
PrP^Sc^ negative (clinical diagnosis at death)	9 ***

* One definite case of Progressive Supranuclear Palsy. ** One definite case of lymphomatosis; *** PrP^Sc^ was negative using Western Blot analysis in the brain samples obtained after autopsy, neuropathological diagnosis is still pending.

**Table 3 ijms-21-00880-t003:** Surrogate biomarkers and real-time quaking-induced conversion (RT-QuIC) testing in CSF and olfactory mucosa (OM) samples of the overall tested subjects.

Biomarker Result	Definite sCJD (*n* = 61)	Probable sCJD (*n* = 41)	Non sCJD (*n* = 80)
14-3-3 +	54 (87%)	33 (80%)	43 (54%)
Tau >1300 pg/mL	56 (92%)	31 (76%)	24 (30%)
CSF RT-QuIC +	58 (95%)	40 (98%)	0
OM RT-QuIC +	16/17 (94%)	16/18 (89%)	0/7
Overall RT-QuIC +	61 (100%)	41 (100%)	0

**Table 4 ijms-21-00880-t004:** Probable and definite sCJD with misleading 14-3-3 and tau testing (in bold are the definite cases).

Patient Code #	14-3-3	Tau (pg/mL)	CSF RT-QuIC	OM RT-QuIC
77	-	266	+	+
80	-	193	+	+
93	-	294	+	+
100	-	373	+	nd
83	+	188	+	nd
**4**	-	1074	+	nd
**14**	-	806	+	nd
**49**	-	1088	+	nd
**52**	-	628	+	nd
**54**	-	1904	+	+
**57**	-	699	+	-
**60**	-	2429	+	nd
64	-	870	+	nd
92	-	1142	+	nd
95	-	562	+	nd
97	-	749	+	-
62	+	1006	+	nd
74	+	1041	+	nd
96	+	1015	+	-

nd—not determined.

**Table 5 ijms-21-00880-t005:** Probable and definite sCJD with negative CSF RT-QuIC testing.

Patient Code #	Duration Month	EEG	MRI	14-3-3	Tau (pg/mL)	CSFRT-QuIC	OMRT-QuIC	Final Clinical DiagnosisCodon 129	Definite DiagnosisPrP^Sc^
13	2	+	+	+	>2400	-	+	Probable sCJDMM	sCJD type1
33	2	+	+	+	>2400	-	+	Probable sCJDMM	sCJD type1
41	2.5	+	+	+	>2400	-	+	Probable sCJDMM	sCJDtype1
81	-	nd	+	+	2101	-	+	Probable sCJDMM

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
