# Peer review of "High Diagnostic Accuracy of RT-QuIC Assay in a Prospective Study of Patients with Suspected sCJD"

_ijms, 2020, doi:10.3390/ijms21030880_

Round 1
Reviewer 1 Report
As I understand it, the principal argument of this paper is that OM RT-QuIC should be considered in patients suspected to have CJD, but in whom the CSF RT-QuIC is negative for purposes of diagnosis, given that they have a combined sensitivity and specificity of 100%. First of all, I should say that there is value in publishing this paper because the scientific method and work here is sound, and it adds to the experience of using OM RT-QuIC in the prion literature; in particular one continues to be impressed with the sensitivity and specificity of the assay in the correct hands and context.
However, it is my impression that, as a clinician who has seen hundreds of CJD cases in person, the the argument here is too focused on the performance of the CSF biomarkers at the expense of the overall clinical picture. From a practical point of view, when you descend into the particulars, OM sampling is perhaps not the path of least resistance in obtaining a secure clinical diagnosis of sCJD even in the cases used for the central argument in this paper. The reasons are below:
When confronted by a patient with an appropriate clinical symptom complex, a supportive MRI is all that it takes to make a clinical (and epidemiological) diagnosis, provided that the CSF does not show high white cell count, and the the CSF routine biochemistry is not wildly abnormal, and potentially reversible causes are excluded. In this context, while a positive RT-QuIC or protein 14.3.3 can be even more reassuring, it is not strictly necessary. Therefore, in my view, the greatest utility of RT-QuIC (CSF or OM) is in those cases where the MRI is atypical, normal, or cannot be obtained for other reasons. However, it appears that all those patients with 'atypical' MRI scans in this paper have positive CSF RT-QuICs anyway, thereby negating the need for OM RT-QuIC. It appears that the 4 patients who had negative CSF RT-QuICs (but positive OM RT-QuICs) had typical clinical features and typical MRIs. This is good enough to achieve a diagnosis of probable sCJD according to the diagnostic criteria referenced in the paper, so why would a clinician want to subject an ill and possibly agitated patient to a nasendoscopy guided OM sampling? Furthermore, the authors showed that by simply diluting the CSF-negative-RT-QuIC samples, positive results were obtained in 2/4 cases in which further CSF is available for testing. So, why not do this routinely before considering getting OM samples. If the MRI is atypical (here defined as cortical ribboning only in one region), why not just repeat the MRI in a week or two, which usually would have evolved into a typical one by then?So, my main criticism of this paper is how the utility of OM RT-QuIC is being portrayed. This should not prevent the paper from being published but perhaps the argument can also consider this logistics of how a patient with suspected CJD is worked up (I am aware that this may vary from country to country) and where OM RT-QuIC might be best used in that context.
A few minor things:
Perhaps the authors might consider using 'disease-associated PrP' (or some other more appropriate term) instead of using "PrPSc" in line 40 (introduction). This can be quite contentious and I know it can mean different things to different people but PrPSc is usually taken to mean PK-resistant PrP which fails to encapsulate all the disease-causing isoforms, a lot of which are probably PK-sensitive. A reference is need for sentence starting with "In fact,..." in line 54. I think the authors meant "sensitivity of about 70%" rather than "specificity of about 70%"? The authors should consider re-writing the sentence starting with "However, RT-QuIC seeding activity...". I think I know what the authors are trying to express but it is not easy to read and understand. I think there is a missing "puncture" after the word "lumbar" in line 234. I would like to know what the two initially negative RT-QuIC CSF samples were diluted in to achieve 1:10 and 1:20 - is it just lower volumes of CSF added per well to make up 100 microlitres or were they diluted in pooled normal human CSF before 20 microlitres of the resultant mix were added? I'm not entirely sure the paragraph starting on line 177 adds anything to the manuscript. AD is very different to CJD clinically and even in the rapid AD cases, the MRI doesn't show CJD features. I think it is meaningless to compare the CSF neurodegenerative markers when the entire clinical picture (including the MRI) is considered.
Reviewer 2 Report
This paper is a follow-up of previous studies on the diagnosis of sporadic CJD based on a cell-free, RT-QuIC assay using CSF and olfactory mucosa brushings from patients. The authors show here the 100% specificity and sensitivity of the assay when performed on both fluids. This provided higher accuracy than any other assays based on CSF and detection of other markers. Such a result is key to the diagnosis of sporadic CJD and differentiation from other rapidly progressive dementia.
The authors should discuss whether the accuracy of such assay may change with the prion strain type causing sporadic CJD cases. Different strains can impact the presence of pathological prion protein in CSF and/or olfactory mucosa. Different types of recombinant PrP may be necessary as a substrate in the RT-QuIC assay to ensure enough sensitivity. Adding more information on the strain types causing the CJD cases included in the study would thus provide useful information.
There are a few typos in the text, e.g. ciclic lane 62.
